# Significance of Indoleamine 2,3-Dioxygenase Expression in the Immunological Response of Kidney Graft Recipients

**DOI:** 10.3390/diagnostics12102353

**Published:** 2022-09-28

**Authors:** Krzysztof Wiśnicki, Piotr Donizy, Agata Remiorz, Dariusz Janczak, Magdalena Krajewska, Mirosław Banasik

**Affiliations:** 1Department of Nephrology and Transplantation Medicine, Wroclaw Medical University, 50-556 Wroclaw, Poland; 2Department of Clinical and Experimental Pathology, Wroclaw Medical University, 50-556 Wroclaw, Poland; 3Department of Vascular, General and Transplantation Surgery, Wroclaw Medical University, 50-556 Wroclaw, Poland

**Keywords:** indoleamine 2,3-dioxygenase (IDO1), immune response, immunosuppression, rejection, immune tolerance, kynurenine pathway, antibody, kidney transplant, graft

## Abstract

Kidney transplantation is unquestionably the most advantageous and preferred treatment when patients with end-stage renal disease are considered. It does have a substantially positive influence on both the quality and expectancy of their lives. Thus, it is quintessential to extend the survival rate of kidney grafts. On account of T-cell-focused treatment, this is being exponentially achieved. The kynurenine pathway, as an immunosuppressive apparatus, and indoleamine 2,3-dioxygenase (IDO1), as its main regulator, are yet to be exhaustively explored. This review presents the recognised role of IDO1 and its influence on the kynurenine pathway, with emphasis on immunosuppression in kidney transplant protection.

## 1. Introduction

Kidney transplantation is the supreme treatment method for patients with end-stage renal disease. It has an essential influence on maximising the lifespan and increasing their quality of life [1]. Various immunosuppressive medications, including those concentrated on T-cell activity, have reduced acute rejection occurrence [2,3,4]. Kidney graft longevity, however, is unsettling.

Indoleamine 2,3-dioxygenase (IDO1) is a cytoplasmic enzyme responsible for inducing and maintaining local immunosuppression. It is associated with almost all of the cells of the immune cell system, such as monocytes, macrophages, and dendritic cells. However, its main activity affects lymphocytes [5,6,7].

There are two paths on which IDO1 can extinguish lymphocytic capacity. Primarily, it is a catabolic enzyme in the first step of the kynurenine pathway of tryptophan, where it is responsible for controlling the rate at which the amino acid is metabolized, causing tryptophan deprivation, thus hindering lymphocytic division [5,6,8,9]. Secondly, the products of the reaction, kynurenines, and their derivatives induce T-cell apoptosis and further inhibit T-cell differentiation [8,9,10].

IDO1 is predominantly activated by interferon γ. [11] However, the expression can also be enhanced by interleukins (IL-1, IL-10) [12,13] and other stimuli, including CD40, superoxide, and lipopolysaccharide anions [14].

This review will focus on IDO1 expression in kidney transplants, its role in reducing graft damage and its influence on both cellular and humoral immune responses.

## 2. Materials and Methods

A literature review was conducted, with editorial comments, letters to the editor, or indexed abstracts at international congresses being omitted. A complex review was executed by comprehensively analysing the literature published up to July 2022. The PubMed, accessed on 31 July 2022, and Google Scholar, accessed on 31 July 2022, databases were searched for variations of keywords: indoleamine 2,3-dioxygenase, IDO1, immune response, immunosuppression, immune tolerance, kynurenine pathway, antibody, kidney transplant, graft, transplantation. Moreover, the PubMed Advanced Search Builder was used, to search for the following phrases: indoleamine 2,3-dioxygenase AND IDO1 OR immune response OR immunosuppression OR immune tolerance OR kynurenine pathway AND kidney graft AND transplant AND lymphocyt* and others.

The literature search was conducted by two authors and about 5400 articles were filtered. We included 90 papers, which specifically concerned indoleamine 2,3-dioxygenase, immunosuppression, and kidney allografts. Abstracts in languages other than English were excluded. Publication review included animal and human studies. Prospective and retrospective clinical studies including center studies, meta-analyses, and review articles were included. Formal institutional review board approval for this study was not required. The search strategy is presented in Figure 1.

## 3. Control of Tryptophan Metabolism

Tryptophan is one of the nine essential amino acids, which cannot be produced by the human organism and must be obtained from the diet [15,16,17]. There are two main metabolism paths of the biomolecule [18]. The minor pathway consists of conversion to serotonin in the nervous system or melatonin production in the conarium. The major route, however, is the kynurenine pathway, which effectively induces nicotinamide adenine dinucleotide generation [15,18,19], an essential coenzyme in metabolism. Importantly, stage products of the kynurenine route also take part in immunosuppression.

The first stage of the kynurenine pathway is the pyrrole ring oxidative cleavage, where tryptophan is metabolized to kynurenine. Two factors can catalyse the reaction, with one being tryptophan 2,3-dioxygenase in the liver [20] and the other being indoleamine 2,3-dioxygenase expressed in various subsystems, tissues, cells [21,22] and, most essentially, in accessory cells which present antigens, also known as antigen-presenting cells [23,24]. In standard circumstances, the liver-allocated metabolism prevails [25,26]. Nevertheless, in the case of immune system activation, the IDO pathway dominates [27,28].

In the inflammatory environment, IDO is activated by interferon-γ, lipopolysaccharides, tumour necrosis factor α, certain interleukins, and transforming growth factor β [29,30]. There are, in fact, two major IDO enzymes that have been emphasised, the aforementioned IDO1 and IDO2. IDO2 is, however, of much less significance and influence. Nonetheless, it is expressed in the liver and kidney [31].

Further steps of the metabolic pathway evolve around kynurenine and its derivatives, such as 3-hydroxykynurenine, anthranilic acid, quinolinic acid, and picolinic acid. The product of the final stage of the kynurenine pathway is the aforementioned NAD+. The kynurenine pathway is presented in Figure 2. Most importantly, however, a study has shown that kynurenine and its metabolites can inhibit T cell proliferative processes through apoptosis [9,18,32]. In summary, IDO does have an immunosuppressive influence on T lymphocytes both by tryptophan depletion and by tryptophan utilisation [18,33].

## 4. Immunosuppressive Mechanisms of IDO1

As mentioned in the introduction, tryptophan catabolism and its influence on the immune system is significant. It has been defined as the tryptophan utilisation theory [18]. The IDO pathway and tryptophan draining lead to direct immune cell inactivation and can deploy regulatory factors [34]. Moreover, tryptophan metabolites, namely, kynurenine, quinolinic and picolinic acids, can directly affect CD8 and CD4 T-helper cells [35].

Different types of immune system cells are affected by or associated with IDO1 in distinct ways. It is a very complex issue involving the whole immune system. To further help understand its significance, several important examples are presented below.

### 4.1. IDO1 and Dendritic Cells

The main role of dendritic cells is antigen presentation and they are crucial in immune system regulation. Their wide range of capabilities consists of uptaking, processing, and presenting antigen, initiating activity of NK and T cells, and, finally, producing cytokines that further affect T and B cell operation. There are two different batches of dendritic cells, namely, conventional DCs and plasmacytoid DCs [23]. Plasmacytoid dendritic cells are responsible for releasing type I and III interferons, interleukins 6, 12, 23, tumour necrotic factor α, as well as class I and II major histocompatibility complex together with CD 40, CD 80, CD 86 proteins, which are pivotal in antigen presentation [24,36]. Going further, by way of illustration, interleukin 12 induces T helper, CD8+ T cell, and NK cell stimulation, parts of the immune system that are very important in viral and intracellular inflammation, while interleukins 6 and 23 induce Th 17 activity, which is decisive in fungal diseases and auto immunological reactions [37] Moreover, plasmacytoid dendritic cells can play a part in tumour necrotic factor apoptosis [38]. In addition, plasmacytoid DCs significance in autoimmune disorders has been noticed, as they are thought to influence B cells directly and then indirectly aid T cell activity [39].

Despite having such a crucial part in pro-inflammatory reactions, dendritic cells can promote immune tolerance, more specifically, on account of IDO1, as they provide essential and IFN-y dependent forms of the enzyme [40,41]. IDO1 is excreted mainly by plasmacytoid DCs [42]. For dendritic cells to express IDO1, aryl hydrocarbon receptor activation is required. Kynurenine and kynurenine pathway products, namely 3-hydroxykynurenine and kynurenic acid, are vital components in this reaction. The process then leads to immune tolerant DCs transformation, and regulatory T cell expression is developed [43,44]. Research has shown the immunomodulatory role of IDO1, where dendritic cells secreting IDO1 suppressed T cell proliferation and induced apoptosis of those cells. Plasmacytoid dendritic cells down-tune the T cell zeta chain receptor and promote regulatory T cell expression.

Moreover, IDO1 is in control of dendritic cell development, migration, and properties. While gaining maturity, DCs relocate to lymphatic organs and their tissues and are capable of inducing immunity.

### 4.2. Macrophages

Macrophages, which belong to the baseline of the innate defence system, can initiate immune reactions. Preceding their divarication, macrophages float in the M0 state. Later, they undergo differentiation into M1 macrophages, which secrete interleukin 12 and tumour necrotic factor α and generally induce inflammation, or M2 macrophages, which produce interleukin 4, interleukin 10 and have restorative properties [45]. Based on their environment and cytokines, macrophages can interconvert between the M1 and M2 types.

Regarding IDO1, its expression in macrophages is possible. However, an IFN-y stimulus is required. Then, the indoleamine 2,3-dioxygenase can switch macrophages from the pro-inflammatory M1 type to their tolerogenic M2 counterparts [21], thus decreasing the immune response.

### 4.3. Natural Killers

Natural Killer cells play a significant role in annihilating external pathogens and neoplastic cells. Abnormal cell division and inflammation lead to NK activation, resulting in the release of cytotoxic granules and death receptor ligands [46]. The aforementioned are two ways NK activity is executed. In the former cytotoxic path, perforin and proteases, also known as granzymes, are secreted. Perforin damages the target cell, and its membrane opens pores, which then allow granzymes to penetrate inside. The latter, the ligand-dependent pathway, is based on death-inducing signalling triggered by ligand binding.

Analysis has shown [47] that IDO mRNA was expressed in NK cells. Providing an IDO1 inhibitor, however, reduced their cytotoxicity. Nevertheless, a different study [48] has shown that IDO1 metabolites are capable of inhibiting NK activation.

### 4.4. Eosinophils

Eosinophils take part in action against parasites and allergic reactions. Multiple cytokines and growth factors originate from eosinophils as well, and are implicated in homeostasis maintenance [49,50]. One of their compelling functions, similar to DCs, is the antigen presentation ability [51].

Moreover, eosinophils can secrete IDO1 [52,53], which then gathers with T cells, leading to inhibition and apoptosis of the Th1 subset [52]. Simultaneously, Th2 expression is enhanced, leading to immune tolerance [54]. In addition, IDO1 secretion has been noticed in thymic eosinophils in newborns [55]. At the same time, tryptophan catabolite, kynurenine, was detected amongst those eosinophils and promotion of Th 2 cells and inhibition of Th1 was confirmed.

### 4.5. Neutrophils

Neutrophils are a crucial factor in events of acute inflammatory reaction and can be dispatched to maligned regions very rapidly. They abolish pathogens in various manners, including phagocytosis, degranulation, and releasing proteolytic enzymes and reactive oxygen species [56].

A particular type of neutrophils that has a significant role in immune response regulation has been found. Myeloid-derived suppressor cells are able to inhibit T cell activity when IDO1 expression is increased [57]. Interleukin 17 enhanced the immunosuppressive action of the aforementioned MDSCs through upregulation of IDO1 [58].

### 4.6. T Cells

There are two main types of T lymphocytes—CD4 expressing T helper cells and CD8 expressing cytotoxic T cells.

Depending on the cytokines they produce, the T helper cells have several subtypes, such as Th1, Th2, Th17, and the regulatory Tregs. Th1 cells, in particular, can release IL-2, IFN-y, and TNF-α and are responsible for macrophage activation [59,60]. Th2 produce interleukins 4,5,9, 10, and 13. They play a significant role in parasitic and allergic reactions. Th17 cells express IL-10, which has anti-inflammatory properties when stimulated with IFN-α or IFN-β [61]. Regulatory T cells, Tregs, have an important influence on tolerance and govern immunotolerance [62]. They produce the transcription factor Foxp3, responsible for their growth, divergence, and regulation [63]. As their name suggests, Tregs play a principal regulatory role in autoimmune disorders.

IDO1 significantly influences T cells, mainly via the tryptophan breakdown, kynurenine pathway, with its metabolites. When insufficient tryptophan is offered to the T cells, they are unable to proliferate after DCs present antigen to them [64]. An amino acid sensor, GCN2K, is activated and inhibits fatty acid production, which in turn halts T cell activation [65,66]. Specifically, CD4 helper cells and Th17 cells have their differentiation suspended, and Th1 cells undergo apoptosis. Moreover, T helper cells can be converted to T regulatory cells instead of being transformed into Th17 cellscite [67]. Thus, anti-inflammatory cytokines are released, and local areas of immunosuppression are generated [18]. The influence of IDO1 on T cells is presented in Figure 3.

### 4.7. B Cells

Up to this point, the vast majority of research has been focused on how IDO1 affects T cells. However, its influence on B cells cannot be ignored. The leading role of B lymphocytes is antibody production. Nevertheless, a subtype of B cells capable of regulating immune response, unconstrained by antibody production, has been found [68]. Specified as regulatory B lymphocytes, Bregs, they impede immune actions using interleukin 10 [69,70]. Moreover, the ability to inhibit Th1 and Th17 differentiation and to enhance antigen presentation by dendritic cells by Bregs has been confirmed [71].

As far as IDO1 is concerned, it has been shown that it can trigger B lymphocytes to become autoreactive at the outbreak of autoimmune reaction. A study has shown that hindering IDO1 results in weakening autoimmune response in arthritis [72].

In opposition, a B cell subtype in mice has been found, which releases IDO imposed by CTLA-4 immunoglobulin stimulation [73]. The CTLA-4 is one of the significant T cell proliferation adjusters that leads to Treg production [74]. However, a study [75] has shown that CTLA-4 can also affect B cells, which in turn generate IDO1, prompt Treg and inhibit Th1 generation. The process is effectuated by TGF-β and IDO1 itself.

## 5. The Role of IDO1 in Kidneys

The role of indoleamine 2,3-dioxygenase in kidney diseases appears to be multifactorial and multithreaded.

In chronic kidney disease, renal fibrosis is the end-stage result of consecutive, lasting cellular and molecular reverberations caused by organ damage. It typically originates from other chronic conditions, predominantly hypertension and diabetes. Fibrosis results in continuous depletion of kidney tissue [76], and for this reason, it is vital to pursue means that can suppress further fibrosis progression. Current treatment methods have been focused on the causes of the phenomenon and can only achieve restricted results as the chronic diseases progress further. Therefore, different ways of treatment are to be explored.

Research [77] has shown that targeted IDO1 inhibition significantly reduced TGF-β-induced fibrosis in kidney cells. During the investigation, two specific IDO1 inhibitors were deployed and reduced gene expression, which resulted in fibrosis attenuation.

In contrast, Volarevic et al. [78] have proven that, in mice kidneys undergoing acute injury, reducing the number of IDO1-expressing dendritic cells, thus lowering kynurenine levels, led to diminished regulatory T cell presence. Moreover, the inflammatory response was boosted as interferon γ and interleukin 17 neutrophils, as well as Th1 and Th17 cells, were congregated. Renal dendritic cells and T regulatory cells relationship has strongly been emphasised as a practical anchor point in acute injury treatment, with IDO1 involved.

Consequently, these results form a strong indication that, in chronic kidney diseases, IDO1-focused therapies should be followed.

## 6. The Role of IDO1 in Kidney Allografts

A different point of view has to be taken into account while considering kidney allografts. Searching for various markers, anchor points, medications, and ways of improving and prolonging graft survival and transplant receiver life expectancy have been and always will be at the focal point of transplantation medicine.

Until now, IDO1 and its activity have been explored mainly in the cellular immune response, evolving around T cells, their division, and how they behave. A study by Cook and colleagues [79] revealed that in kidney allografts in mice, IDO gene expression incrementally grew over time. Regulatory T cells and IDO-affected dendritic cells were detected in the non-rejected transplants, which strongly indicated their role in allograft tolerance. In addition, as the aforementioned regulatory dendritic cells were identified in spleens of allograft recipient mice, their donor-matched skin transplants were deemed accepted. Therefore, allograft acceptance should be considered a progressive arrangement and a result of a series of different procedures.

Moreover, during research performed on rats by Na et al. [80], an anti-IDO lentivirus was created. Its properties aimed at interfering with IDO expression in dendritic cells, which, as expected, promoted T cell proliferation in the mixed leukocyte reaction. Furthermore, pre-immunization tests with donor alloantigens, specifically with alloantigen-affected immature dendritic cells, were executed and revealed inhibited rejection reaction in comparison to non-pre-immunized allografts. The study strongly indicated that new methods of graft survival rate extension, remarkably involving IDO and dendritic cells, should be pursued. In addition, in an inquiry by Brandacher et al. [81], urine, blood, and tissue samples were assembled from patients with and without acute graft rejection. The study has shown that during episodes of acute rejection, the kynurenine-to-tryptophan ratio in urine and serum was elevated. The ratio then translated directly into IDO activity, as confirmed in immunostaining on renal biopsy materials. Additionally, intensified IDO expression in renal tubules was detected. At the same time, in healthy, non-rejected kidney transplants, IDO activity was not observed.

Nevertheless, the humoral, antibody-mediated immune response cannot be ignored when IDO1 is considered. As mentioned above, it is crucial to pursue new ways of protecting transplanted organs. As a matter of fact, antibody-mediated rejection is the leading player in graft immunological injury and transplant loss, with the massive significance of human leukocyte antigen and the anti-HLA and non-HLA antibodies [82,83,84,85,86,87,88]. Recent studies have shown that especially the non-HLA antibodies, including anti-ETAR antibodies, should be pursued, as their importance is being revealed [89]. When considering IDO1, some studies [90] have shown promise in its influence on the humoral response.

## 7. Conclusions

In summary, the role of indoleamine 2,3-dioxygenase in organ transplantation should be emphasized and further investigated. It is a critical approach and direction in prolonging the allograft survival rate. Managing the kynurenine pathway, thus hindering T cell response, adjusting the operation of dendritic cells, and prompting T regulatory cell reaction presents a promising outlook in clinical transplantation. Up to this point, the aforementioned research has been performed in vitro or in animals. However, with sufficient amounts of data, the effect of IDO1 should be explored in humans.

What should be the subject of the upcoming studies is, however, the impact of IDO1 on B cells and the antibody-mediated response, especially when taking donor-specific antibodies into consideration [89]. Perhaps indoleamine-2,3-dioxygenase will be revealed as a significant factor in this area.

Perchance, in the future, IDO1-inducing treatment may appear, either in monotherapy or combined with existing immunosuppressive therapies.

## Figures and Tables

**Figure 1 diagnostics-12-02353-f001:**
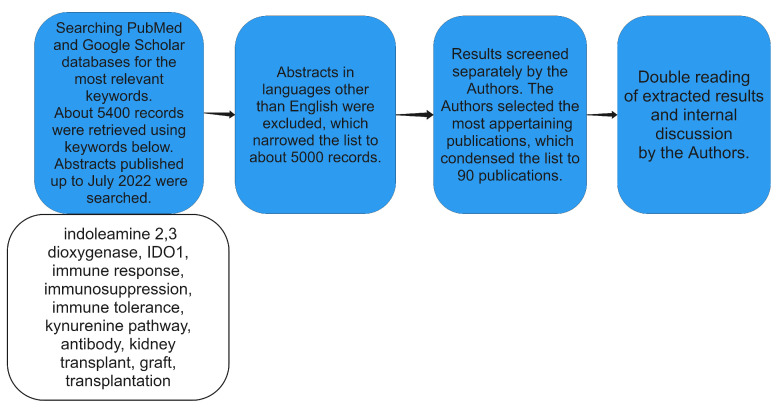
Strategy of data collection presented as a flow diagram.

**Figure 2 diagnostics-12-02353-f002:**
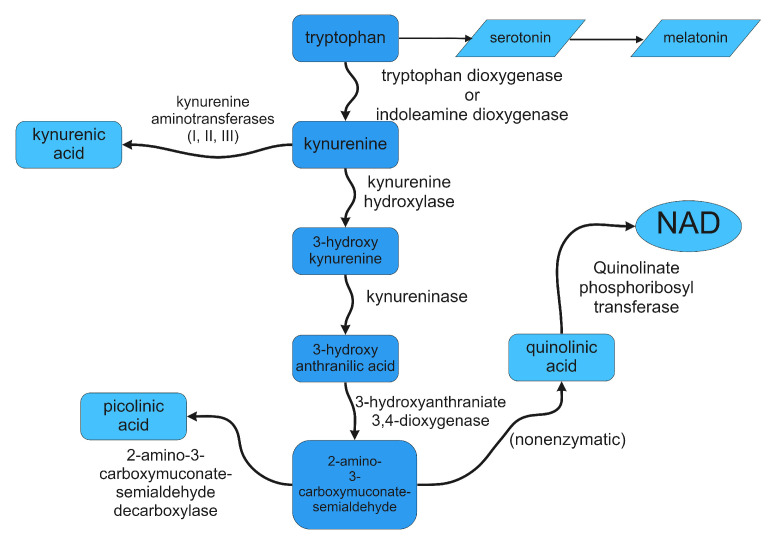
The kynurenine pathway of tryptophan.

**Figure 3 diagnostics-12-02353-f003:**
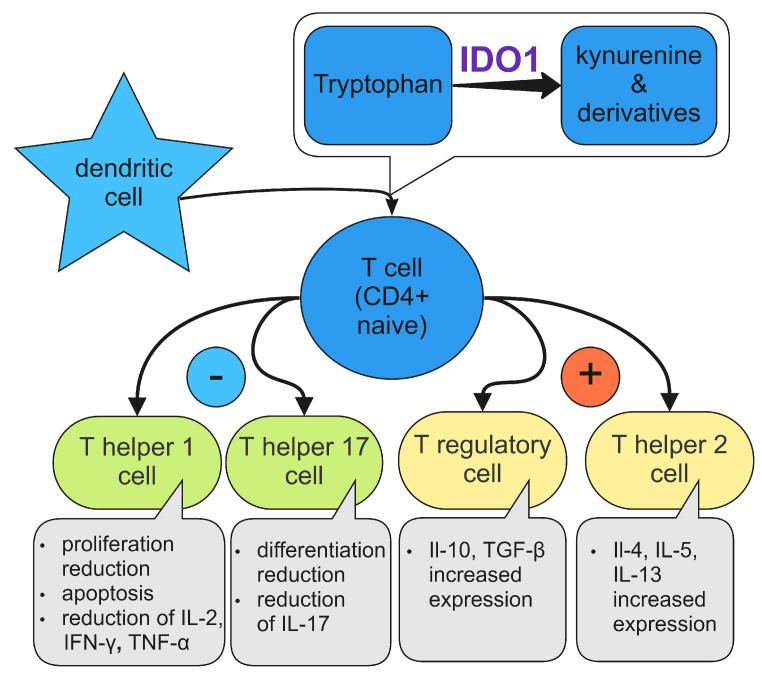
The influence of IDO1 on T cells.

## Data Availability

Not applicable.

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
