# Peer review of "Significance of Indoleamine 2,3-Dioxygenase Expression in the Immunological Response of Kidney Graft Recipients"

_diagnostics, 2022, doi:10.3390/diagnostics12102353_

Round 1
Reviewer 1 Report
The review article titled ‘Significance of indoleamine 2-3 dioxygenase expression in the immunological response of kidney graft recipients’ summarized the influence of indoleamine 2,3-dioxygenase (IDO1) on the tryptophan metabolism pathway, kynurenine pathway, and emphasis on immunosuppression in kidney transplant protection. The authors first discussed around the involvement of IDO enzymes in the tryptophan metabolism pathway, with the initiation of kynurenine metabolism and the final metabolites of picolinic acid, quinolinic acid and NAD+. Then the immunosuppressive mechanisms of IDO1 were discussed thoroughly, which were refined into seven aspects, includes dendritic cells, macrophages, natural killers, eosinophils, neutrophils, T cells and B cells. To better illustrating how IDO1 can functioning in the kidney allografts, the role of IDO1 in kidney was discussed. Finally, using case study and clinical study result to prove the importance of IDO1 in kidney graft recipients.
Overall, I think this article is well organized and written, and the summary is comprehensive. Although minor things need to be improved, such as font in the graph is unclear and graph quality, these are not hindering the good quality of this review paper. I suggest accepting this article with minor revision.
Author Response
Dear Reviewer,
Please see the attachment. The response has been uploaded in a Word file.
Kind regards,
Krzysztof Wiśnicki

Reviewer 2 Report
Dear Authors,
I have read the manuscript and I send you my comments:
1) methods must be added, how to reviewed the data?
2) Clinical data must be added, several references are related to experimental animal studies, so human data are necessary.
3) please add a table resuming the data in the text
Author Response
Dear Reviewer,
Please see the attachment. Our response has been uploaded in a Word file.
Kind regards,
Krzysztof Wiśnicki

Round 2
Reviewer 2 Report
Dear authors I have read the revised version and I have not other comments
Best regards